# Characterization of Four Rearing Managements and Their Influence on Carcass and Meat Qualities in Charolais Heifers

**DOI:** 10.3390/foods11091262

**Published:** 2022-04-27

**Authors:** Julien Soulat, Brigitte Picard, Cécile Bord, Valérie Monteils

**Affiliations:** 1Institut National de Recherche pour l’agriculture, l’alimentation et l’environnement (INRAE), VetAgro Sup, Unité Mixte de Recherche sur les Herbivores (UMR Herbivores), Université Clermont Auvergne, F-63122 Saint-Genès-Champanelle, France; julien.soulat@inrae.fr (J.S.); valerie.monteils@vetagro-sup.fr (V.M.); 2INRAE, VetAgro Sup, Unité Mixte de Recherche sur le Fromage (UMR Fromage), Université Clermont Auvergne, F-63370 Lempdes, France; cecile.bord@vetagro-sup.fr

**Keywords:** carcass traits, meat traits, hierarchical clustering, multifactorial, sensory qualities, rearing factors, rearing surveys, whole life

## Abstract

The study aim was to identify the effects of the rearing management applied throughout the heifers’ life on the carcass (e.g., conformation, marbling, fat) and meat (color, texture, and sensory profiles) properties. From the individual data of 171 heifers from 25 commercial farms, a typology of four rearing managements was established from 50 rearing factors. The managements had an effect on the conformation, the color (fat and muscle), and the *rhomboideus* grain meat, for the carcass, and the lightness, the atypical flavor, and the overall acceptability for the *longissimus* (LM) meat. The carcass traits compared to the meat were more sensitive to a change of rearing management. Our results confirmed that it was possible to target the same carcass or meat quality from different managements. Moreover, according to the aims of the targeted carcass and LM meat quality, management 3 could be an interesting trade-off to jointly manage the quality of both products. For example, the carcasses that were produced had a high conformation, smooth meat grain and the LM meat was more liked. This management was intermediate compared to the other rearing managements and had a long fattening period with a diet mainly based on conserved grass and a high concentrate quantity.

## 1. Introduction

In the European Union, the beef production between 2016 and 2020 was relatively stable [1]. In 2020, the meat production in the European Union represented 17.6% of the world production. France is the first producer of beef in the European Union with a relatively stable production since 2017 [1]. An important part of this production comes from systems based on grass-fed and valorized by different official quality labels [2]. In addition, the beef cuts sold (butcher shop or hyper and supermarkets) mainly come from the suckling herd (42% cull cows and 64% heifers) [3].

However, for many years, the beef sector has had to adapt to produce carcasses and beef corresponding to market and societal expectations. For different reasons (e.g., price, environment, animal welfare, rearing management, health), beef consumption has decreased [4,5]. The trend of the consumers in the European Union and in France is to reduce the quantity and increase the quality of the beef consumed [4]. For the next decade, the challenge of the beef sector is to increase beef quality increasing the share of products that have labels (e.g., “Label Rouge”, Protected Denomination of Origin, Protected Geographical Indication) or are organic, related to specifications [4,6]. The sensory properties (e.g., flavor, tenderness, color, juiciness) of the beef stay factors in determining consumer satisfaction and their decision to purchase it [4,7,8].

The beef quality is impacted by many factors, e.g., rearing factors [9,10,11], carcass and muscles traits [12,13,14,15], aging [16,17], or cooking temperature [18,19]. However, in Europe, the breeders’ income is based on the carcass traits, especially the weight, conformation, and fat scores [20,21]. As for meat quality, the cattle rearing management throughout their life (from birth to slaughter) impact the carcass quality [12,22]. In this study, rearing management is defined as a combination of many rearing factors (e.g., weaning age, fattening duration, pasture duration, concentrate quantity) characterizing the breeding strategy followed by the heifers from their birth to their slaughter.

Consequently, early and joint management of carcass and beef qualities are relevant to study to meet beef stakeholders’ and the consumers’ expectations.

The aims of this study were to characterize rearing managements followed by the heifers during their life, then, to identify the effects of these managements simultaneously on carcass and meat traits.

## 2. Materials and Methods

In our study, the animals were reared and slaughtered in commercial farms and industrial slaughterhouse. The method used did not have any impacts on the rearing managements applied by farmers during the animals’ life and the slaughter conditions used in the slaughterhouse. Consequently, as the animals were reared and slaughtered for production of French meat production, a validation by a research ethics committee was not necessary to perform this study according the French legislation. 

### 2.1. Animals and Rearing Factors

The individual data of 171 Charolais heifers from 25 French commercial farms in three departments (Allier, Loire, and Puy-de-Dôme) of the Auvergne-Rhône-Alpes region, a main production area in France, were included in this study. The farms were chosen to have a high diversity of rearing managements within this area. The heifers were born between August 2015 and October 2018, and slaughtered between May 2019 and October 2020. 

A single survey was performed for each farm and allowed to collect data for 50 rearing factors (Table 1, Table 2 and Table 3) characterizing three key periods of the whole heifers’ life (from birth to slaughter): pre-weaning period (PWP), growth period (GP), and fattening period (FP). Each survey (around 2 h) was carried out by interviewing the farmers using questionnaires (open-ended and closed-ended questions) and by establishing batch management practices [23]. During the survey, the composition (forages and concentrates) of each diet distributed to heifers during their life (from birth to slaughter), the housing entry and exit dates, the dates of weaning, beginning at fattening, and slaughter were collected on the batch management practices. From the batch management practices established, the different rearing factors were calculated for each heifer.

The composition and the nutritional values of each purchased concentrates were collected from the manufactures. For the concentrate produced on the farm, e.g., barley or wheat, the nutritional values of the INRAE system were used [24]. As described by Soulat et al. [10], from the nutritional values of the concentrates, the average concentrate’s crude protein (CP) and the average concentrate’s net energy (NE) contents were calculated for the whole outside period, for the whole housing period, and for the whole of each key period of the heifer’s life (PWP, GP, and FP, respectively). The average percentage of forages (e.g., hay, grass silage, corn silage, etc.) was calculated for the average housing diet of the whole growth period and for the whole fattening period. 

### 2.2. Animals Slaughtering, Carcass Traits Assessment and Sampling

The 171 heifers were slaughtered in four French industrial slaughterhouses (Puigrenier, Monluçon, France; SICABA, Bourbon l’Archanbault, France; SICAREV, Roanne, France, and SOCOPA, Villefranche-d’Allier, France). The slaughter was performed in compliance with European regulation No 1099/2009 on the protection of animals at the time of killing [25].

The heifers were stunned using a captive-bolt pistol prior to exsanguination and the carcasses were not electrically stimulated. The carcasses were suspended vertically using the Achilles method. At the end of the slaughtering line, carcasses were weighed and graded visually by an official judge (conformation and fat scores) according to the EUROP grid system [21]. After, the carcasses were chilled and stored at 2 °C until 24 h *post-mortem*.

The carcass conformation was graded according to a scale of 15 subclasses: E+ (very high muscle development) to P- (very low muscle development). A numerical conversion was conducted for statistical analysis with a scale between 1 (corresponding P-) and 15 (corresponding E+). The fat score was graded according to a scale of five classes (1 = lean to 5 = very fat) (Table A1). 

The carcasses were cut at the 6th rib level 24 h *post-mortem*. Then, 11 carcass traits were assessed visually or by touch by trained slaughterhouse staff (operator). 

The subcutaneous fat thickness assessment was performed in the area presented in Figure 1 using a caliper. The seepage of *longissimus* muscle (LM), intermuscular fat, nerves, overall meat grain, *longissimus* meat grain, and *rhomboideus* (RH) meat grain (Table 4 and Table A1) were assessed with a scale from 1 to 5 with a step of 0.5 [26]. The homogeneous color of muscles was assessed with a scale including four modalities according to the number of distinguished colors (Table 4). The fat color, the color, and marbling of LM, were assessed with the color charts and the marbling scale described by UNECE [27]. The distribution of these carcass traits were presented in the Table A1.

After that, a sample of two beef ribs (the 5th and the 4th ribs) was collected and deboned. Each sample was individually vacuum-packaged and aged for 14 days at 4 °C. Then, the meat samples were frozen at −20 °C until the analyses. The beef rib is composed of different muscles [11]. To characterize the sample collected, the analyses on the meat were focused on two muscles: LM for color, texture, and sensory analyses, because it is the most studied for meat qualities and the *serratus ventralis* muscle (SV), for shear force analysis because it is a specific muscle in the ribs of the chuck sale section.

### 2.3. Meat Quality Assessment

The beef rib samples were thawed for around 48 h at +4 °C. Then, for each beef rib sample, the LM and the SV were dissected by professional butchers (INRAE Unité Expérimentale Herbipôle, Theix, France). Each muscle was individually vacuum packaged. The LM samples were used to perform the color, sensory, and texture analyses. The SV samples were frozen at −20 °C until the shear force analyses.

#### 2.3.1. Color Assessment

The color of the LM samples was measured three hours before the sensory analyses, using a spectrophotometer (Konica Minolta CR-400, Osaka, Japan) and expressed in CIE L*a*b* units [28]. The tests were performed by applying the parameters: D65 illuminant and observer angle of 2°. The color measure was directly performed on the surface of the LM sample on the same area of LM as the visual color assessment in slaughterhouse. To characterize the color of the whole LM sample, the mean of six measurements (randomly distributed on the muscle) per LM sample, was used. As described by Soulat et al. [22], the spectrophotometer used in this study was calibrated following the instruction of the manufacturer.

#### 2.3.2. Sensory Analysis

Fifty people were recruited to participate in this study according to their sensory abilities. A training session was conducted including six 1-h sessions. During this step, they were trained to recognize different perceptions and use the perception scales according to ISO 8586 [29]. In this study, the sensory grid was composed of 10 sensory descriptors and one hedonic descriptor. These descriptors are defined in Table 5. Afterward, panel performances (consensus, discrimination, and repeatability) were evaluated and verified. For each evaluation session, 10-trained panelists were selected. The sensory descriptor assessment was performed using a 10-cm unstructured scale (from 0 = no perception to 10 = perception very intense). For overall acceptability, a scale anchoring from “0 = I don’t like at all to 10—I like very much” was used.

From each LM sample, two or three steaks with a 2 cm thickness were cut. The steaks were placed in aluminum foil and were cooked in a plancha at 300 °C to reach an internal endpoint cooking temperature of 55 °C. After cooking, the steaks were cut into homogeneous pieces (size 15 × 20 × 20 mm) and kept warm. For each sensory session, eight samples were evaluated monadically using a Latin square design. In this study, 87 sensory sessions were performed. All tests took place in individual sensory booths in accordance with ISO 8589 [30]. Mineral water (Evian, France) and unsalted crackers were served to rinse the mouth during the test. For each sample, the individual assessment was performed on a sensory survey using the Tastel software^®^ (ABT Informatique, Rouvroy-sur-Marne, France). 

The rest of each LM sample was individually vacuum-packaged and frozen at −20 °C until the texture analyses.

#### 2.3.3. Texture Profile Analysis

These samples were thawed for 25 min and cut using a cookie cutter to obtain a regular cylinder (1 cm thick and 1 cm in diameter). The meat cylinders were placed at +4 °C to finish their thawing (around 30 min). When it was not possible to perform at least two meat cylinders, the sample was not analyzed. Each sample underwent two cycles of 20% compression at 4 °C using rheometer (Kinexus pro+, Malvern Instruments, Malvern, UK) and the rSpace 1.61 software (Kinexus, Malvern, UK). The force-deformation curve, obtained during the texture profile analysis test, allowed to calculate six parameters: springiness, hardness, cohesiveness, resilience, gumminess, and chewiness [31,32].

#### 2.3.4. Shear Force Measurement

The shear force (N/cm^2^) was measured on raw meat for all SV samples. These samples were thawed at +4 °C for around 24 h. For each sample, around 14 meat portions (1 cm wide and between 0.9 and 1.1 cm thick) were cut in parallel to the fibers. Two shear force measurements (cut perpendicular to the fibers) were performed by meat portion using the Warner-Braztler method (Instron 5944, Elancourt, France) and the Bluehill 2 software (Instron, Elancourt, France). Around 20 shear force measurements per SV samples were performed. Then, the mean of these measurements was used.

### 2.4. Statistical Analyses

Statistical analyses were performed using R 4.0.5 software [33].

For each rearing factor, a descriptive analysis was performed (e.g., graphic distribution and quantile-quantile plots). After this descriptive analysis, some quantitative rearing factors were converted into qualitative rearing factors, as described by Soulat et al. [10].

The rearing managements applied throughout the life of the heifers were defined from all rearing factors (q = 50, presented in Table 1, Table 2 and Table 3) using the factor analysis for mixed data (FAMD) followed by a hierarchical clustering on principal components (HCPC). The dendrogram of the HCPC allowed to determine the number of rearing managements to be considered in this study. The FAMD and HCPC were performed using “FactoMineR” package in R [34].

Then, between the defined rearing managements, ANOVAs and chi-squared tests were carried out for each rearing factor (quantitative and qualitative, respectively) to evaluate their dependence on the defined rearing managements. Moreover, when the result of the ANOVA was significant (*p* ≤ 0.05), a Tukey test was performed.

For each carcass and meat trait considered in this study, an ANOVA was also performed to evaluate the effect of the rearing management on each carcass or meat traits. 

For carcass traits, the slaughterhouse and the operator effects were considered as random effects in mixed models if they were significant (*p* ≤ 0.05). For the sensory data, the panelist was considered as random effect in mixed models.

ANOVAs were performed using “agricolae” package [35] and the mixed models followed by Tukey test using “lmerTest” [36], “emmeans” [37], “multcompView” [38], and “multcomp” [39].

## 3. Results and Discussion

### 3.1. Description of the Rearing Managements Defined

Four rearing managements (RM) applied during the heifers’ whole life were defined by hierarchical clustering on principal components.

The first rearing management (RM-1) was performed on 44 heifers. The main characteristics of RM-1 compared to the other managements were:

There was the highest percentage of artificial insemination (Table 1). During PWP, the calves had the longest pasture duration and the shortest period with concentrates in their diet. The calves did not receive concentrates in pasture. At housing, 70.5% and 79.5% of the calves did not receive forages and concentrates in their diet, respectively. The heifers were weaned at the oldest age. 

The heifers had the longest GP duration and the longest outside and pasture durations during this period (Table 2). Outside, the heifers received forage complementation during a longer period than the heifers from RM-2 and RM-4. In their housing diet, the heifers had a higher percentage of hay than RM-2 and RM-3, and the lowest percentage of grass silage. Moreover, 90.9% of the heifers did not receive corn silage in their housing diet, and 43.2% received above 50% of wrapped haylage. During GP, the heifers received concentrates for the shortest period and ingested the lowest concentrate quantity. The average concentrate consumed by the heifers during this period had the lowest CP and NE contents. Outside, 97.7% of the heifers did not receive concentrates.

During FP, the heifers were the oldest at the beginning of the fattening and at slaughter (Table 3). The FP duration was shorter than RM-2 and RM-3, and 70.5% of the heifers were fattened in pasture. In RM-1, for 61.4% of the heifers, the pasture duration was below 100 days, and 45.5% of the heifers were fattened with only grass as fiber source. During FP, the heifers consumed the lowest concentrate quantity. The average concentrate consumed by the heifers during this period had the lowest CP and NE contents. 

The second rearing management (RM-2) was performed by 59 heifers. The main characteristics of RM-2 compared to the other managements were:

The calves received forages in their diet during a longer period than the calves from RM-4, and 62.7% of the calves received forages during the housing period (Table 1). The calves received concentrate in their diet during a longer period than the calves from RM-1 and RM-3, and 62.7% of the calves received concentrate during the pasture. During PWP in housing, a high proportion of the calves (52.5%) received an average concentrate with a CP content lower than the calves from RM-3 and RM-4. However, in pasture, the majority of the calves received the average concentrate with the highest CP content. During the housing period and the whole PWP, the majority of the calves received an average concentrate with the highest NE content.

The heifers had the shortest GP and a shorter pasture period than RM-1 and RM-3 (Table 2). During the housing period, the heifers received on average a lower proportion of hay and a higher proportion of grass silage than the heifers from RM-1 and RM-4. Moreover, 64.4% of the heifers did not receive wrapped haylage and 50.8% received below 25% of corn silage. The duration of concentrate distribution in the heifers’ diet was intermediate compared to the other managements, throughout GP. Moreover, the heifers ingested a lower concentrate quantity than the heifers from RM-3 and RM-4. During the housing period, the average concentrate intake had a higher CP content than RM-1 and RM-3. During the pasture, 81.4% of the heifers did not receive concentrates. 

In RM-2, the heifers were the youngest at the beginning of FP and were slaughtered younger than those from RM-1 and RM-3 (Table 3). In RM-2, the heifers’ fattening duration was longer than in RM-1 and RM-4, and 54.2% of the heifers were fattened in a stall. However, 44.1% of the heifers had a period above 100 days during their fattening. In RM-2, the majority of heifers (44.1%) were fattened with corn silage as main conserved forage in their diet. The heifers consumed a higher concentrate quantity than those from RM-1 and RM-4. The average concentrate intake had a high CP and NE contents.

The third rearing management (RM-3) was performed by 32 heifers. The main characteristics of RM-3 compared to the other managements were:

The calves (78.1%) received forages in their diet in stall (Table 1). Throughout PWP, the concentrate distribution was shorter compared to RM-2 and RM-3. At pasture, 68.8% of the calves did not receive concentrate. During PWP in housing, a high proportion of the calves (78.1%) received an average concentrate with a CP content higher than the calves from RM-1 and RM-2.

The GP duration of the heifers was intermediate compared to the other managements. Similarly, the outside and pasture durations were also intermediate (Table 2). During the housing period, the heifers received a lower proportion of hay than the heifers from RM-1 and RM-4. Although 50% of the heifers did not receive corn silage, 31.2% received the highest corn silage percentage (between 25% and 40%). Moreover, 43.8% of the heifers received below 50% of wrapped haylage. At pasture, the heifers were complemented with forages during a higher duration than heifers from RM-2 and RM-4. Throughout GP, the heifers ingested a higher concentrate quantity than those from RM-1 and RM-2. During the outside period, 53.1% of the heifers consumed above 150 kg of concentrates. The average concentrate intake during GP had high CP and NE contents. In RM-3, all heifers received concentrates in pasture and 68.8% received an average concentrate with the highest NE content.

The heifers were slaughtered older than the heifers from RM-2 and RM-4 (Table 3). The heifers started the fattening with an intermediate age compared to the other RM. They had a longer fattening duration than the heifers from RM-1 and RM-2. The fattening was mainly performed in a stall (62.4% of the heifers) and the diet can be compounded by different main forages. The heifers consumed the highest concentrate quantity. For FP, the average concentrate had contents of CP and NE, intermediate and high, respectively.

The fourth rearing management (RM-4) was performed by 36 heifers. The main characteristics of RM-4 compared to the other managements were:

The majority of the calves (>63%) did not have a complementation with conserved forages, throughout PWP (Table 1). Moreover, the calves received concentrate during the longest period. All calves received concentrate in pasture. Throughout PWP, the majority of calves (66.7%) received an average concentrate with a high CP content (>17%) and a NE content below 7.5 kJ.

The heifers had an intermediate GP duration compared to the other RM but they had the longest period in a stall (Table 2). In the housing diet, the heifers received a higher percentage of hay than heifers from RM-2 and RM-3. Moreover, the heifers did not receive wrapped haylage and the majority of heifers (61.1%) received corn silage in this diet. The heifers received concentrates during the longest period and consumed the highest quantity. At pasture, 44.5% of the heifers did not receive concentrates and 47.2% received above 150 kg. During the housing period, the average concentrate had a higher CP content than in RM-1 and RM-3. Outside, all heifers receiving concentrate consumed an average concentrate with an NE below 7.5 kJ.

The heifers were slaughtered younger than the heifers from RM-1 and RM-3 (Table 3). The fattening duration was shorter than this in RM-2 and RM-3. The fattening was performed mainly in a stall. The heifers were mainly fattened with diets based on corn silage or straw. The heifers ingested a lower concentrate quantity than the heifers from RM-2 and RM-3. During FP, the average concentrate had high contents of CP and NE.

The main characteristics of the four rearing managements are summarized in Figure 2. In summary, the heifers performing RM-1 had a long duration outside during their life and consumed few concentrates. These heifers were weaned, fattened, and slaughtered older. In the RM-2, the heifers had a short GP, and they were fattened and slaughtered younger. These heifers were mainly fattened in housing; however, some heifers had a period on pasture during their fattening. The RM-3 was intermediate compared to the three other RM. The heifers had a longer FP with diet based mainly on conserved grass (hay, silage, wrapped haylage) and a high concentrate quantity. The heifers performing the RM-4 had a long duration in housing and ingested a high concentrate quantity throughout their life. These heifers were slaughtered young with a short fattening duration based on straw or corn silage.

### 3.2. Effects of the Rearing Managements on the Carcass Traits

According to our results, the RM influenced carcass traits related to conformation and color (e.g., fat color, homogeneous color of the cut section, and LM color) (Table 4). In the literature, few works studied the effect of cattle’s whole life on the carcass properties.

Carcasses with higher conformation scores were obtained when the heifers performed the RM-2 compared to those performing RM-1. However, the carcass conformations of RM-2, RM-3, and RM-4 were not significantly different (*p* > 0.05). Our results are in accordance with Soulat et al. [12,22] showing that RM with the longest pasture duration during the heifers’ whole life as RM-1 (Figure 2) produced carcasses with low conformation. However, for these authors similar conformation could be reached with different RM. As previously observed, the combination of different rearing practices [22] or rearing factors [40] applied throughout the heifers’ life can allow to produce carcasses with a similar weight, conformation, and/or dressing.

At the 6th rib level, the LM and overall meat grains were not significantly impacted by RM (Table 4). The meat grain was mainly related to the muscle properties, in particular the muscle fiber size [41]. To our knowledge, the effect of RM on the meat grain has not been studied. However, the effect of some individual rearing factors on the fiber size was observed [42]. The slaughter age above 19 months [43], the fattening diet [44], and the physical activity [45] had no effect on the LM fiber size, in young bulls. However, the RH had a grain meat rougher for carcasses from heifers performing the RM-2 compared to carcasses from heifers performing the RM-1. The RH grain meat was not significantly different between the RM-1 and the managements RM-3 and RM-4. Compared to the LM, the RH muscle have a higher proportion of type slow I fibers and a lower proportion of fast type II fibers [46]. The RH muscle is a postural muscle. It is possible that RM had an effect on the fiber size of this muscle. To our knowledge, it is the first time that the effect of RM was studied on the grain meat of this muscle. This result will have to be confirmed by other results.

The color of the cut section was significantly more homogeneous for heifers performing the RM-2 than those performing the RM-4 (Table 4). At 24 h *post-mortem*, the LM was significantly darker for heifers from the RM-1 than those from the RM-2. In the RM-1, the heifers had a longer pasture duration than those performing the RM-2. In accordance with this result, many studies showed that the meat redness (lower a* value) was lower for the young bulls fattened in pasture than those fattened in stall [47,48]. However, it was difficult to compare our results with previous published works because they were mostly carried out on FP and observe the individual effect of rearing factor. Nevertheless, Soulat et al. [49] also showed that the heifers performing an RM with a long pasture duration and fattening in pasture produced a darker LM meat than those performing an RM with a fattening only in housing. Cozzi et al. [50] observed an opposite effect in heifers. The LM color was not significantly different between, RM-2, RM-3, and RM-4 (Table 4). In these three RM, although the heifers were mainly fattened in housing, some heifers were fattened in pasture or in pasture and housing. Moreover, the fattening diets were different between the three RM. As observed in this study, many studies did not observe a significant effect of the fattening diet on the LM redness (a* parameter) [16,51]. Other studies showed that the LM redness can also be similar to RM with different fattening systems (pasture, housing, pasture and housing, and outside) [12,49]. Concerning the fat color at the cut section level, heifers from the RM-1 produced carcasses with yellower fat than those from RM-2 and RM-4. The fattening diet in the RM-2 and RM-4 was mainly based on corn silage or straw (Table 3). The fat color of carcasses was not significantly different between RM-1 and RM-3. In RM-3, 46.9% of the heifers received a fattening diet based only on grass-conserved forages (hay, grass silage, and/or wrapped haylage). Our results are in accordance with those of Velik et al. [52] and Duckett et al. [53] who observed that the fat was yellower when heifers or steers were fattened in pasture compared to those fattened in housing. Moreover, Varela et al. [54] and French et al. [55] observed that the subcutaneous fat was yellower when the cattle ingested grass in fattening diet compared to those consuming fattening diet based on corn silage plus concentrate, or grass silage, or hay. Contrary to these authors, the fat color was not significantly different when the heifers were fattened in pasture or with fattened diet based on grass-conserved forages, in our study. According to our results, RM did not significantly affect the other carcass traits related to fat (Table 4).

According to our results, RM had no significant effect on the weight, the LM seepage, the LM proportion, and the nerve abundance (Table 4).

Our results must be confirmed by others because there are few results that consider heifers and the whole life of cattle. Consequently, it is difficult to compare our results with other published results.

According to the carcass quality target, considering joint carcass traits, different RM could be prioritized. From RM-1 and RM-3, the carcass traits were not significantly different (Table 4). The effects of RM-2 and RM-3 on the carcass traits were not significantly different. The management RM-2 allowed the production of carcasses with high conformation and a wither fat than the RM-1. According to our results, it is difficult to favor an RM. The significant difference for each trait was generally only between two RM. It is necessary to carry out trade-off according to the quality target.

### 3.3. Effects of the Rearing Managements on the Meat Traits

According to our results, the RM had few effects on the analyzed meat traits (Table 6). These results are in accordance with Soulat et al. [12,22,49] in heifers.

After aging, only the L* color parameter of raw LM meat was significantly different between the four RM and tendencies were observed for both other color parameters (Table 6). The LM meat from heifers performing the RM-2, was lighter compared to RM-1 and RM-3. In the RM-2, the heifers were slaughtered younger than those from RM-1 and RM-3. Moreover, in the RM-2, most of heifers were fattened in housing compared to those from the RM-1 that were fattened in pasture. Other studies showed that the LM meat aged 14 days had a higher L* values when the animals (steers or young bulls) were fattened in housing than those fattened in pasture [47,56]. In heifers, Cozzi et al. [50] obtained the same results with LM meat aged 8 days. As for our results on the carcass, it is difficult to compare our results because there are few published results considering the heifer’s whole life. The heifers performing the RM-4 had the tendency to produce a redder raw meat than the other RM. The heifers performing the RM-2 had the tendency to produce a raw meat with a lower b* value than the other RM. It was possible that the effect of RM observed at 24 h *post-mortem* on LM meat color (Table 4) was mitigated by ageing. Some studies observed an effect of the aging on the meat color [16,57]. However, Moloney et al. [57] did not observe an effect of the interaction between aging and fattening growth rate strategy.

The toughness of the LM (measured by texture profile analysis) and SV (measured by shear force) raw meats were not significantly different between RM (Table 6) as observed in previous results of Soulat et al. [11,49], in heifers. According to the results of the texture profile analysis, RM did not influence texture traits of the LM raw meat (Table 5). In accordance with our results, Marino et al. [58] did not observe an effect of the fattening system (pasture vs. housing) on the texture profile of raw meat, in young bulls. However, Garcia-Torres et al. [59] observed that the cooked LM was significantly tougher when the young bulls were fattened in pasture than those in housing pens.

In accordance with the results about raw meat toughness, the tenderness of the cooked LM meat evaluated by the trained panel did not display significant differences between RM (Table 6). Among the 10 sensory descriptors studied, only the atypical flavor (Table 5) of LM meat was significantly different between RM. It was higher when the heifers performed the RM-1 compared to those performing the RM-4. Soulat et al. [49] observed also an effect of RM on the typical flavor of LM meat, in heifers. Concerning the sensory descriptors related to the texture (e.g., initial and overall tenderness, overall juiciness), no significant effect of RM were observed in previous results of Soulat et al. [12,49]. However, in previous studies, Soulat et al. [12] did not show a significant effect of RM observed on the overall juiciness on the mean of two muscles LM and *rectus abdominis*. The heifers performing RM-1 and RM-2 had the tendency to produce an LM meat with more nerves than those performing RM-3 and RM-4 without consequence on the tenderness of raw and cooked LM meat (Table 6). This tendency could be explained by the pasture effect on the connective tissue. The pasture duration was more important in the RM-1 and RM-2 than the RM-3 and RM-4. Duckett et al. [53] and Serrano et al. [60] observed an increase of the total collagen content or of the soluble collagen/total collagen ratio, respectively, when grass was included in the fattening system.

Regarding the overall acceptability (hedonic descriptor), the heifers performing the RM-4 with a lower pasture duration produced an LM meat, which was appreciated significantly more by the trained panel than those performing RM-1 and RM-2 (Table 6). In a previous study, a trained panel appreciated the LM meat more from heifers performing the fattening without pasture than those having a long pasture duration during only FP or during the heifers’ whole life [12].

To produce the highest overall meat quality (considered the whole traits), the RM-3 and RM-4 would be prioritized. These RM allowed the production of LM meat with less lightness (weak L* value), with weak atypical flavor and highly appreciated, as the other meat traits were not impacted by RM (Table 6).

Our results showed that RM can have little influence on LM meat properties (e.g., lightness, atypical flavor, and overall acceptability) and also that different RM can produce LM meat with similar properties (e.g., tenderness, juiciness, and flavor intensity). As observed for the carcass, our results need to be confirmed by others because there are few results on meat quality in heifers and considering the whole life of cattle.

To jointly manage carcass and meat properties from RM, the RM-3 allows an interesting trade-off to produce carcasses and LM meat with the highest overall qualities. RM-3 allowed the production of carcasses with a high conformation and smooth meat grain. The rib section was more homogeneous, the fat was yellower, and the LM was redder. RM-3 was characterized as being intermediate of the other RM (Figure 2). In RM-3, the heifers received a fattening diet based on conserved grass forages with a high concentrate quantity during a long period.

## 4. Conclusions

The originality of this study was to show the impact of RM applied throughout the heifers’ life on both carcass and meat qualities. From the individual data of 171 heifers on their breeding in different commercial farms, four RM were defined statistically. These RM were established from 50 rearing factors of the three main heifer’s life periods.

Our results confirmed that the RM had an effect on the carcass and meat traits. Consideration of the heifer’s whole life with combinations of many rearing factors allows to establish a relationship with product qualities. Moreover, our results also confirmed that the carcass traits were more sensitive than the meat properties when the rearing management was changed. However, it was difficult to favor an RM to obtain the best carcass quality. It was necessary to perform trade-offs according to the quality target. Regarding the LM meat traits, the sensory properties (except L*, atypical flavor, and overall acceptability) were very similar whatever the RM. This diversity of RM confirmed that it is possible to maintain or obtain similar properties in terms of carcass and meat qualities. Moreover, these results confirmed that it was possible to jointly manage the overall carcass and LM meat qualities from RM. According to our results, the management allowing the best trade-off between carcass and meat qualities was an intermediate rearing management with a long fattening period whose diet was mainly based on conserved grass and a high concentrate quantity. This management allowed the production of (i) carcasses with high conformation, smooth meat grain, a rib section with a more homogeneous color, a darker LM, and a yellower fat and (ii) LM meat with low lightness and atypical flavor, and more liked. To further complete this study, it would be interesting to integrate the production cost to more precisely characterize RM.

## Figures and Tables

**Figure 1 foods-11-01262-f001:**
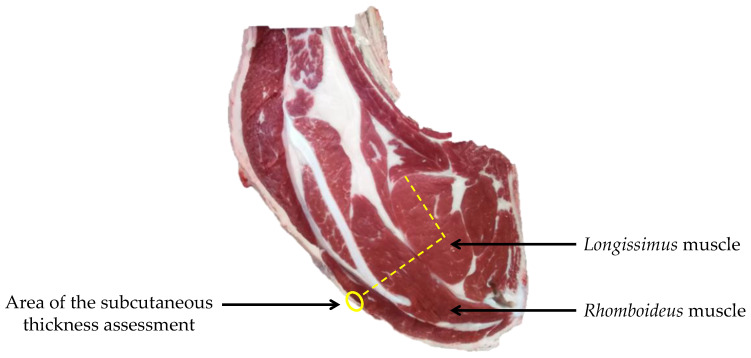
Picture at the 6th rib level and localization of some muscles and measures.

**Figure 2 foods-11-01262-f002:**
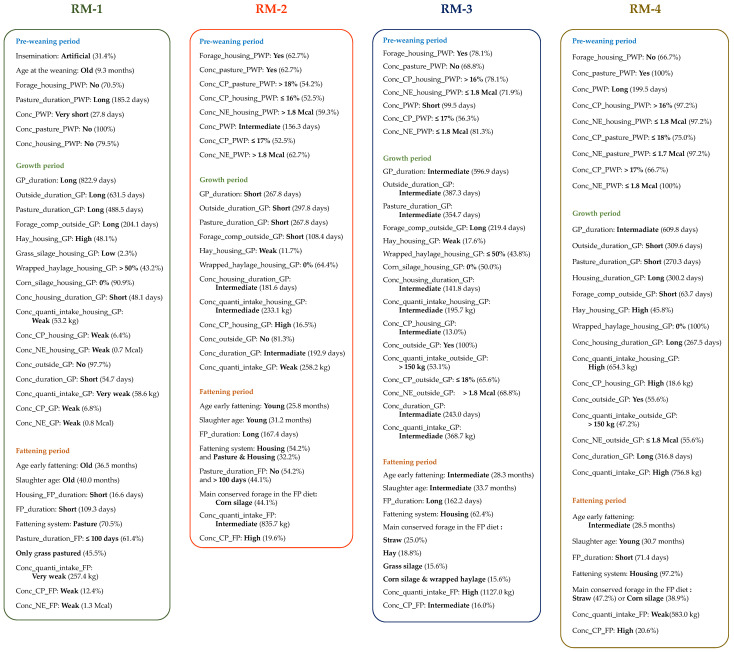
Summary of the four rearing managements (RM) applied during the whole life of the heifers. PWP: Pre-weaning period, GP: Growth period, FP: Fattening period, Conc_CP: Calculated average of concentrate’s crude protein, Conc_NE: Calculated average of concentrate’s net energy, Conc: Concentrate, Conc_quanti_intake: Total concentrate quantity intake per heifer, Forage_comp: Number of days when forages were offered outside, Main forage: Main forage in the fattening diet.

**Table 1 foods-11-01262-t001:** Rearing factors characterizing the pre-weaning period (PWP) of the rearing managements (RM) applied during the heifers’ whole life.

Rearing Factors		Description of the Rearing Factor	Overall		Rearing Managements								*p*
					RM-1		RM-2		RM-3		RM-4		
			*n* = 171		*n* = 44		*n* = 59		*n* = 32		*n* = 36		
Quantitative Rearing Factors			Mean	SE	Mean	SE	Mean	SE	Mean	SE	Mean	SE	
Age of the cow (year)		Age of the heifer’s mother at the heifer’s birth	5.9	0.2	5.9	0.4	5.5	0.3	6.5	0.4	6.3	0.5	0.28
Age at the first calving (year)		Age of the heifer’s mother at first calving	3.0	0.01	3.0	0.03	3.0	0.01	3.0	0.02	3.0	0.03	0.21
Age at the weaning (month)		Age of heifer at the weaning	8.6	0.09	9.3 ^a^	0.2	8.3 ^b^	0.1	8.3 ^b^	0.2	8.5 ^b^	0.1	<0.001
Housing_duration_PWP (day)		Numbers of days spent in stall during PWP	100.3	3.9	98.6	7.9	91.9	8.4	99.9	6.5	116.8	4.5	0.14
Pasture_duration_PWP (day)		Number of days spent in pasture during PWP	160.1	3.6	185.2 ^a^	5.9	157.5 ^b^	8.1	153.2 ^b^	5.5	139.9 ^b^	2.7	<0.001
Tot_forage_duration_PWP (day)		Number of days of offered forages in the calves’ diet during PWP	79.1	5.5	66.2 ^ab^	10.6	99.8 ^a^	9.1	80.7 ^ab^	10.2	59.4 ^b^	13.6	0.03
Conc_housing_PWP (day)		Number of days of offered concentrates in the calves’ diet during housing	55.8	4.1	27.8 ^c^	8.3	52.9 ^bc^	7.3	64.1 ^ab^	6.5	87.4 ^a^	6.6	<0.001
Conc_PWP (day)		Number of days of offered concentrates in the calves’ diet during PWP	121.7	7.6	27.8 ^d^	8.3	156.3 ^b^	13.9	99.5 ^c^	12.3	199.5 ^a^	6.5	<0.001
**Qualitative Rearing Factors**	**Modalities**												
Insemination type	Artificial	Artificial insemination using frozen semen	17.5%		34.1%		11.9%		15.6%		8.3%		0.01
	Natural	Insemination performed by a bull	82.5%		65.9%		88.1%		84.4%		91.7%		
Calving	Easy	Natural calving	78.9%		27.3%		15.3%		28.1%		16.7%		0.31
	Help	Farmer intervention during the calving	21.1%		72.7%		84.7%		71.9%		83.3%		
Forage_housing_PWP	Yes	Offered forages in housing calves’ diet during PWP	50.9%		29.5%		62.7%		78.1%		33.3%		<0.001
	No	No offered forages in housing calves’ diet during PWP	49.1%		70.5%		37.3%		21.9%		66.7%		
Forage_pasture_PWP	Yes	Offered forages in pasture calves’ diet during PWP	30.4%		27.3%		33.9%		21.9%		36.1%		0.53
	No	No offered forages in pasture calves’ diet during PWP	69.6%		72.7%		66.1%		78.1%		63.9%		
Conc_pasture_PWP	Yes	Offered concentrates in pasture calves’ diet during PWP	48.5%		0%		62.7%		31.3%		100%		<0.001
	No	No offered concentrates in pasture calves’ diet during PWP	51.5%		100%		37.3%		68.8%		0%		
Conc_CP_housing_PWP (%)	No	No offered concentrates in housing calves’ diet during PWP	35.1%		79.5%		35.6%		9.4%		2.8%		<0.001
	≤16%	Across the whole housing of PWP, the calculated average of concentrate’s crude protein content was below 16%	25.7%		20.4%		52.5%		12.5%		0%		
	>16%	Across the whole housing of PWP, the calculated average of concentrate’s crude protein content was above 16%	39.2%		0%		11.9%		78.1%		97.2%		
Conc_NE_housing_PWP (KJ)	No	No offered concentrates in housing calves’ diet during PWP	35.1%		79.5%		35.6%		9.4%		2.8%		<0.001
	≤7.5 kJ	Across the whole housing of PWP, the calculated average of concentrate’s net energy content was below 7.5 kJ	35.7%		0%		5.1%		71.9%		97.2%		
	>7.5 kJ	Across the whole housing of PWP, the calculated average of concentrate’s net energy content was above 7.5 kJ	29.2%		20.5%		59.3%		18.8%		0%		
Conc_CP_pasture_PWP (%)	No	No offered concentrates in pasture calves’ diet during PWP	51.5%		100%		37.3%		68.8%		0%		<0.001
	≤18%	Across the whole pasture of PWP, the calculated average of concentrate’s crude protein content was below 18%	23.9%		0%		8.5%		28.1%		75.0%		
	>18%	Across the whole pasture of PWP, the calculated average of concentrate’s crude protein content was above 18%	24.6%		0%		54.2%		3.1%		25.0%		
Conc_NE_pasture_PWP (KJ)	No	No offered concentrates in pasture calves’ diet during PWP	51.5%		100%		37.3%		68.8%		0%		<0.001
	≤7.1 kJ	Across the whole pasture of PWP, the calculated average of concentrate’s net energy content was below 7.1 kJ	31.0%		0%		30.5%		0%		97.2%		
	>7.1 kJ	Across the whole pasture of PWP, the calculated average of concentrate’s net energy content was above 7.1 kJ	17.5%		0%		32.2%		31.3%		2.8%		
Conc_CP_PWP (%)	No	No offered concentrates during PWP	26.4%		79.5%		13.6%		6.3%		0%		<0.001
	≤17%	Across the whole PWP, the calculated average of concentrate’s crude protein content was below 17%	40.9%		20.4%		52.5%		56.3%		33.3%		
	>17%	Across the whole PWP, the calculated average of concentrate’s crude protein content was above 17%	32.7%		0%		33.9%		37.5%		66.7%		
Conc_NE_PWP (KJ)	No	No offered concentrates diet during PWP	26.4%		79.5%		13.6%		6.3%		0%		<0.001
	≤7.5 kJ	Across the whole PWP, the calculated average of concentrate’s net energy content was below 7.5 kJ	44.4%		20.5%		23.7%		81.3%		100%		
	>7.5 kJ	Across the whole PWP, the calculated average of concentrate’s net energy content was above 7.5 kJ	29.2%		0%		62.7%		12.5%		0%		

*n*: number of heifers. SE: Standard error. Values followed by different letters (a, b, c) are significantly different from each other at *p* ≤ 0.05.

**Table 2 foods-11-01262-t002:** Rearing factors characterizing the growth period (GP) of the rearing managements (RM) applied during the heifers’ whole life.

Rearing factors		Description of the Rearing Factor	Overall		Rearing Managements								*p*
					RM-1		RM-2		RM-3		RM-4		
			*n* = 171		*n* = 44	*n* = 59	*n* = 32		*n* = 36				
Quantitative Rearing Factors			Mean	SE	Mean	SE	Mean	SE	Mean	SE	Mean	SE	
Housing_duration_GP (day)		Number of days spent in stall during GP	230.5	5.8	191.4 ^c^	12.1	228.4 ^b^	9.7	209.6 ^bc^	10.2	300.2 ^a^	5.0	<0.001
Outside_duration_GP (day)		Number of days spent outside during GP	402.9	14.0	631.5 ^a^	27.6	297.8 ^c^	13.7	387.3 ^b^	13.5	309.6 ^c^	10.9	<0.001
Pasture_duration_GP (day)		Number of days spent in pasture during GP (heifers graze)	341.4	10.9	488.5 ^a^	26.0	267.8 ^c^	9.3	354.7 ^b^	16.6	270.3 ^c^	9.5	<0.001
GP_duration (day)		Number of days between the weaning and the beginning of the fattening	633.4	12.7	822.9 ^a^	23.9	526.2 ^c^	15.6	596.9 ^b^	16.9	609.8 ^b^	10.4	<0.001
Forage_comp_outside_GP (day)		Number of days when forages were offered during the whole outside period of GP	144.4	7.9	204.1 ^a^	15.8	108.4 ^b^	10.5	219.4 ^a^	16.5	63.7 ^b^	5.9	<0.001
Hay_housing_GP (%)		Calculation of the hay percentage in the average housing diet across the whole GP	29.4	2.2	48.5 ^a^	5.0	11.7 ^b^	2.5	17.6 ^b^	3.1	45.8 ^a^	3.4	<0.001
Grass_silage_housing_GP (%)		Calculation of the grass silage percentage in the average housing diet across the whole GP	36.4	2.4	2.3 ^c^	1.3	56.4 ^a^	3.5	43.8 ^ab^	5.7	38.8 ^b^	2.7	<0.001
Conc_housing_duration_GP (day)		Number of days of offered concentrates in the housing diet during GP	157.9	8.9	48.1 ^c^	10.3	181.6 ^b^	13.4	141.8 ^b^	16.7	267.5 ^a^	12.7	<0.001
Conc_duration_GP (day)		Number of days of offered concentrates in the diet during GP	192.8	10.1	54.7 ^c^	11.6	192.9 ^b^	15.3	243.0 ^b^	17.7	316.8 ^a^	9.8	<0.001
Conc_quanti_intake_housing_GP (kg)		Total concentrate quantity intake per heifer during the housing period	268.5	21.9	53.2 ^c^	11.2	233.1 ^b^	20.4	195.7 ^b^	23.0	654.3 ^a^	57.4	<0.001
Conc_quanti_intake_GP (kg)		Total concentrate quantity intake per heifer during the whole GP	332.5	23.3	58.6 ^d^	11.9	258.2 ^c^	22.6	368.7 ^b^	35.0	756.8 ^a^	43.3	<0.001
Conc_CP_housing_GP (%)		Calculated average of concentrate’s crude protein content across the whole housing period	13.7	0.6	6.4 ^c^	1.3	16.5 ^a^	0.5	13.0 ^b^	1.4	18.6 ^a^	0.4	<0.001
Conc_NE_housing_GP (kJ		Calculated average of concentrate’s net energy content across the whole housing period	6.4	0.2	0.7 ^c^	0.1	1.9 ^a^	0.01	1.6 ^b^	0.2	1.8 ^ab^	0.02	<0.001
Conc_CP_GP (%)		Calculated average of concentrate’s crude protein content across the whole GP	14.4	0.5	6.8 ^b^	1.3	16.5 ^a^	0.5	16.5 ^a^	0.6	18.3 ^a^	0.2	<0.001
Conc_NE_GP (kJ)		Calculated average of concentrate’s net energy content across the whole GP	6.7	0.2	0.8 ^b^	0.1	1.9 ^a^	0.01	1.9 ^a^	0.03	1.8 ^a^	0.02	<0.001
**Qualitative Rearing Factors**	**Modalities**												
Wrapped_haylage_housing_GP (%)	0%	Across the GP, the heifers had no wrapped haylage in the housing diet	57.9%		29.5%		64.4%		37.5%		100%		<0.001
	≤50%	Across the GP, the calculated average percentage of wrapped haylage in the housing diet was below 50%	23.4%		27.3%		23.7%		43.8%		0%		
	>50%	Across the GP, the calculated average percentage of wrapped haylage in the housing diet was above 50%	18.7%		43.2%		11.9%		18.8%		0%		
Corn_silage_housing_GP (%)	0%	Across the GP, the heifers had no corn silage in the housing diet	56.1%		90.9%		44.1%		50.0%		38.9%		<0.001
	<25%	Across the GP, the calculated average percentage of corn silage in the housing diet was below 25%	31.0%		6.8%		50.8%		18.8%		38.9%		
	[25%; 40%]	Across the GP, the calculated average percentage of corn silage in the housing diet was between 25% and 40%	12.9%		2.3%		5.1%		31.2%		22.2%		
Conc_outside_GP	Yes	Offered concentrates during the outside period	37.4%		2.3%		18.6%		100%		55.6%		<0.001
	No	No offered concentrates during the outside period	62.6%		97.7%		81.3%		0%		44.4%		
Conc_quanti_intake_outside_GP (kg)	0 kg	No offered concentrates during the outside period	62.6%		97.7%		81.4%		0%		44.5%		<0.001
	≤150 kg	Total concentrate quantity intake per heifer during the outside period was above 150 kg	16.9%		0%		18.6%		46.9%		8.3%		
	>150 kg	Total concentrate quantity intake per heifer during the outside period was below 150 kg	20.5%		2.3%		0%		53.1%		47.2%		
Conc_CP_outside_GP (%)	No	No offered concentrates outside	62.6%		97.7%		81.4%		0%		44.5%		<0.001
	≤18%	Across the outside period, the calculated average of concentrate’s crude protein content was below 18%	16.9%		0%		0%		65.6%		22.2%		
	>18%	Across the outside period, the calculated average of concentrate’s crude protein content was above 18%	20.5%		2.3%		18.6%		34.4%		33.3%		
Conc_NE_outisde_GP (kJ)	No	No offered concentrates outside	62.6%		97.7%		81.4%		0%		44.4%		<0.001
	≤7.5 kJ	Across the outside period, the calculated average of concentrate’s net energy content was below 7.5 kJ	24.6%		2.3%		18.6%		31.3%		55.6%		
	>7.5 kJ	Across the outside period, the calculated average of concentrate’s net energy content was above 7.5 kJ	12.8%		0%		0%		68.8%		0%		

*n*: number of heifers. SE: Standard error. Values followed by different letters (a, b, c, d) are significantly different from each other at *p* ≤ 0.05.

**Table 3 foods-11-01262-t003:** Rearing factors characterizing the fattening period (FP) of the rearing managements (RM) applied during the heifers’ whole life.

Rearing Factors		Description of the Rearing Factor	Overall		Rearing Managements								*p*
					RM-1		RM-2		RM-3		RM-4		
			*n* = 171		*n* = 44		*n* = 59		*n* = 32		*n* = 36		
Quantitative Rearing Factors			Mean	SE	Mean	SE	Mean	SE	Mean	SE	Mean	SE	
Age early fattening (month)		Age of the heifer at the beginning of FP	29.6	0.4	36.5 ^a^	0.7	25.8 ^c^	0.5	28.3 ^b^	2.0	28.5 ^b^	0.4	<0.001
Slaughter age (month)		Age of the heifer at the slaughter	33.8	0.4	40.0 ^a^	0.6	31.2 ^c^	0.7	33.7 ^b^	0.5	30.7 ^c^	0.5	<0.001
Housing_duration_FP (day)		Number of days spent in stall during the FP	71.3	5.7	16.6 ^b^	4.3	99.1 ^a^	10.2	99.7 ^a^	18.3	67.2 ^a^	4.9	<0.001
FP_duration (day)		Number of days between the beginning of FP and the slaughter	131.3	5.8	109.3 ^b^	7.3	167.4 ^a^	10.1	162.2 ^a^	14.4	71.4 ^b^	7.3	<0.001
Conc_quanti_intake_FP (kg)		Total concentrate quantity intake per heifer during the whole FP	688.2	41.9	257.4 ^d^	34.0	835.7 ^b^	65.1	1127.0 ^a^	115.3	583.0 ^c^	64.1	<0.001
Conc_CP_FP (%)		Calculated average of concentrate’s crude protein content across the whole FP	17.3	0.5	12.4 ^c^	1.2	19.6 ^a^	0.5	16.0 ^b^	0.4	20.6 ^a^	0.7	<0.001
Conc_NE_FP (kJ)		Calculated average of concentrate’s net energy content across the whole FP	7.1	0.2	1.3 ^b^	0.1	1.9 ^a^	2.3	1.9 ^a^	0.02	1.8 ^a^	0.03	<0.001
**Qualitative Rearing Factors**	**Modalities**												
Fattening system	Housing	The fattening was carried out in stall	52.6%		6.8%		54.2%		62.4%		97.2%		<0.001
	Pasture and Housing	The fattening started in pasture and was finished in stall	18.7%		22.7%		32.2%		6.3%		2.8%		
	Pasture	The fattening was carried out in pasture	28.7%		70.5%		13.6%		31.3%		0%		
Pasture_duration_FP (day)	No pasture	No pasture during the FP	53.2%		9.1%		54.2%		62.5%		97.2%		<0.001
	≤100 days	During the FP, the number of days in pasture was below 100 days	18.2%		61.4%		1.7%		9.4%		0%		
	>100 days	During the FP, the number of days in pasture was above 100 days	28.6%		29.5%		44.1%		28.1%		2.8%		
Main conserved forage in the FP diet (%)	Grass_silage_and_wrapped_haylage_FP	The percentage of the sum of grass silage and wrapped haylage in the FP diet was above 85%	1.7%		0%		1.7%		6.2%		0%		<0.001
	Corn_silage_FP	The percentage of corn silage in the FP diet was above 90%	24.6%		4.5%		44.1%		0%		38.9%		
	Grass_silage_FP	The percentage of grass silage in the FP diet was above 90%	6.4%		0%		1.7%		15.6%		13.9%		
	Hay_FP	The percentage of hay in the FP diet was above 95% (except for one animal, hay = 64%)	9.4%		0%		16.9%		18.8%		0%		
	Straw_FP	The percentage of straw in the FP diet was above 75%	16.4%		0%		5.1%		25.0%		47.2%		
	Wrapped_haylage_FP	The percentage of wrapped haylage in the FP diet was above 70%	8.2%		20.5%		5.1%		6.3%		0%		
	Corn_silag_and_wrapped_haylage_FP	The percentage of the sum of corn silage and wrapped haylage in the FP diet was above 80%	6.4%		2.3%		8.5%		15.6%		0%		
	Hay_and_wrapped_haylage_FP	The percentage of the sum of hay and wrapped haylage in the FP diet equal 100%	8.8%		27.3%		5.1%		0%		0%		
	No	No offered conserved forages in the FP diet	18.1%		45.5%		11.9%		12.5%		0%		

*n*: number of heifers. SE: Standard error. Values followed by different letters (a, b, c, d) are significantly different from each other at *p* ≤ 0.05.

**Table 4 foods-11-01262-t004:** Effects of the four rearing managements (RM) applied during the whole heifers’ life factors on the carcass traits.

Carcass Traits	Description of the Carcass Trait	Overall		Rearing Mangements								*p*
				RM-1		RM-2		RM-3		RM-4		
		*n* = 171		*n* = 44		*n* = 59		*n* = 32		*n* = 36		
		Emmean	SE	Emmean	SE	Emmean	SE	Emmean	SE	Emmean	SE	
Cold weight (kg)		403	10	386	12	406	12	392	13	392	13	0.12
Conformation score (scale 1 to 15)	EUROP classification scale for conformation (from P− = 1 to E+ = 15)	9.0	0.1	8.6 ^b^	0.2	9.2 ^a^	0.2	8.9 ^ab^	0.2	9.0 ^ab^	0.2	0.003
Fat score (scale 1 to 5)	EUROP classification scale for fat score (1 = lean to 5 = very fat)	3.0	0.05	2.9	0.1	3.0	0.0	2.9	0.04	3.0	0.1	0.14
**Assessment at the 6th rib level**		***n* = 137**		***n* = 43**		***n* = 36**		***n* = 22**		***n* = 36**		
*Longissimus* muscle seepage (scale 1 to 5)	*Longissimus* muscle seepage assessment (1 = the cut section is dry with no drop to 5 = the cut section has important drop)	1.8	0.2	1.8	0.3	2.2	0.3	2.0	0.3	1.7	0.3	0.12
Subcutaneous fat (cm)	Measure of the subcutaneous fat thickness	0.8	0.2	0.7	0.4	1.5	0.4	0.8	0.4	0.8	0.4	0.12
Inter-muscular fat (scale 1 to 5)	Inter-muscular fat assessment (1 = limited development to 5 = large amount)	2.1	0.2	2.0	0.3	2.3	0.3	2.0	0.3	2.1	0.3	0.63
Nerves (scale 1 to 5)	Nerves assessment (1 = lack of visible nerves to 5 = many visible nerves)	1.3	0.1	1.4	0.2	1.4	0.2	1.5	0.2	1.4	0.2	0.92
Overall meat grain (scale 1 to 5)	Overall meat grain assessment (1 = smooth, soft, without harshness to 5 = very rough/granular)	1.9	0.2	2.0	0.2	2.3	0.23	2.1	0.2	2.0	0.2	0.27
*Longissimus* meat grain (scale 1 to 5)	*Longissimus* meat grain assessment by touch (1 = smooth, soft, without harshness to 5 = very rough/granular)	1.6	0.2	1.5	0.3	1.9	0.3	1.7	0.3	1.8	0.3	0.15
*Rhomboideus* meat grain (scale 1 to 5)	*Rhomboideus* meat grain assessment by touch (1 = smooth, soft, without harshness to 5 = very rough/granular)	1.3	0.2	1.2 ^a^	0.2	1.7 ^b^	0.2	1.4 ^ab^	0.2	1.4 ^ab^	0.2	0.02
Fat color (scale 0 to 9)	Fat color assessment using the color chart described by UNECE [27]	2.6	0.3	3.2 ^a^	0.4	2.5 ^bc^	0.4	2.7 ^ac^	0.4	2.1 ^b^	0.4	<0.001
Homogeneous color of muscles at the 6th rib (scale 1 to 4)	Homogeneous color assessment between muscles (1 = homogeneous, 2 = bicolor, 3 = tricolor, and 4 = more than three colors)	1.8	0.07	1.8 ^ab^	0.1	1.6 ^b^	0.1	1.7 ^ab^	0.1	2.0 ^a^	0.1	0.05
*Longissimus* color (scale 0 to 7)	*Longissimus* muscle color assessment using the color chart described by UNECE [27]	4.2	0.3	4.7 ^a^	0.4	4.6 ^ab^	0.4	3.9 ^ab^	0.4	3.7 ^b^	0.4	0.04
*Longissimus* marbling (scale 0 to 6)	Longissimus marbling assessment using the marbling scale described by UNECE [27]	1.5	0.1	1.3	0.2	1.5	0.3	1.6	0.3	1.6	0.2	0.49

Emmean: estimated marginal means. *N*: number of heifers. SE: Standard error. Values followed by different letters (a, b, c) are significantly different from each other at *p* ≤ 0.05.

**Table 5 foods-11-01262-t005:** Definitions of the sensory and hedonic descriptors.

Descriptors	Definition
Red color intensity	Refers to the red color intensity of the meat sample after cooking (0 = light to 10 = dark)
Initial tenderness	Facility to chew and cut the meat sample at the first bite (0 = tough to 10 = very tender)
Overall tenderness	Time and numbers of chewing required to masticate the meat sample ready for swallowing (0 = tough to 10 = very tender)
Overall juiciness	Perception of water content in the meat sample during the mastication (0 = dry to 10 = very juicy)
Presence of nerves	Quantities of nerves perceived in the meat sample (0 = none to 10 = very important)
Residue	Amount of the residue after chewing (0 = none to 10 = very important)
Flavor intensity	Global flavor intensity assessment of the beef (0 = none to 10 = very intense)
Fat aroma	Fat aroma intensity (0 = none to 10 = very intense)
Atypical flavor	Flavor associated with aromas that should not normally be present in meat (e.g., aftertaste, rancid) (0 = none to 10 = very intense)
Flavor persistence	Refers to remnant beef flavor duration in the mouth perceived after swallowing (0 = very quick to 10 = very long)
Overall acceptability	Overall liking (hedonic perception) of the meat sample (0 = highly disliked to 10 = highly liked)

**Table 6 foods-11-01262-t006:** Effects of the four rearing managements (RM) applied during the whole heifers’ life factors on the meat traits.

Meat Traits	Overall		Rearing Managements								*p*
			RM-1		RM-2		RM-3		RM-4		
	Mean	SE	Mean	SE	Mean	SE	Mean	SE	Mean	SE	
**Raw Meat**											
*Longissimus* muscle	***n* = 152**		***n* = 36**		***n* = 52**		***n* = 30**		***n* = 34**		
Texture profile analysis											
Springiness	0.5	0.01	0.5	0.01	0.5	0.01	0.4	0.01	0.5	0.01	0.06
Hardness (N)	1.6	0.05	1.6	0.1	1.6	0.1	1.7	0.1	1.8	0.1	0.52
Cohesiveness	2.2	0.1	2.6	0.3	2.0	0.1	2.2	0.2	2.1	0.2	0.15
Resilience	0.2	0.01	0.3	0.02	0.2	0.01	0.2	0.02	0.2	0.02	0.17
Gumminess	3.6	0.2	4.1	0.5	3.3	0.3	3.7	0.4	3.6	0.4	0.46
Chewiness	1.7	0.09	1.9	0.2	1.6	0.1	1.7	0.2	1.8	0.2	0.59
Color descriptors	***n* = 163**		***n* = 43**		***n* = 52**		***n* = 32**		***n* = 36**		
L*	41.9	0.2	41.1 ^b^	0.5	42.9 ^a^	0.4	41.1 ^b^	0.5	42.2 ^ab^	0.4	0.01
a*	18.9	0.3	18.4	0.7	18.4	0.4	18.6	0.8	20.3	0.5	0.10
b*	12.6	0.1	12.9	0.2	12.1	0.2	12.6	0.3	12.8	0.2	0.07
*Seratus ventralis* muscle	***n* = 162**		***n* = 42**		***n* = 52**		***n* = 32**		***n* = 36**		
Shear force (N/cm^2^)	66.6	1.4	66.2	3.1	65.0	2.5	66.26	2.5	70.0	3.4	0.65
**Cooked Meat**											
*Longissimus* muscle	***n* = 155**		***n* = 38**		***n* = 52**		***n* = 29**		***n* = 36**		
	**Emmean**	**SE**	**Emmean**	**SE**	**Emmean**	**SE**	**Emmean**	**SE**	**Emmean**	**SE**	
Sensory descriptors (0–10 scale) ^1^											
Red color intensity	3.5	0.2	3.8	0.2	3.4	0.2	3.7	0.3	3.4	0.3	0.23
Initial tenderness	6.4	0.1	6.3	0.2	6.3	0.2	6.5	0.2	6.4	0.2	0.59
Overall tenderness	6.0	0.2	5.8	0.2	5.9	0.2	6.1	0.2	6.1	0.2	0.28
Overall juiciness	4.6	0.2	4.6	0.2	4.6	0.2	4.5	0.2	4.8	0.2	0.60
Presence of nerves	1.9	0.2	2.0	0.2	2.0	0.2	1.8	0.2	1.8	0.2	0.06
Residue	3.1	0.2	3.2	0.2	3.2	0.2	3.3	0.2	3.2	0.2	0.96
Flavor intensity	5.9	0.1	5.9	0.1	5.8	0.1	5.9	0.2	5.8	0.2	0.75
Fat aroma	3.7	0.2	3.6	0.2	3.7	0.2	3.5	0.3	3.8	0.2	0.14
Atypical flavor	0.8	0.2	1.0 ^a^	0.2	0.75 ^ab^	0.2	0.8 ^ab^	0.2	0.6 ^b^	0.2	<0.001
Flavor persistence	4.9	0.2	4.9	0.2	4.8	0.2	4.9	0.2	4.9	0.2	0.82
Overall acceptability	5.8	0.1	5.3 ^c^	0.2	5.7 ^bc^	0.2	5.9 ^ab^	0.2	6.2 ^a^	0.2	<0.001

Emmean: estimated marginal means. *n*: number of heifers. SE: Standard error. Values followed by different letters (a, b, c) are significantly different from each other at *p* ≤ 0.05. ^1^, The sensory descriptors were described in Table 5.

## Data Availability

No new data were created or analyzed in this study. Data sharing is not applicable to this article.

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
