# Peer review of "Characterization of Four Rearing Managements and Their Influence on Carcass and Meat Qualities in Charolais Heifers"

_foods, 2022, doi:10.3390/foods11091262_

Round 1

Reviewer 1 Report

Report on paper no: foods-1669562

Influence of the Rearing Managements on the Qualities of Carcass and Beef Meat in Charolais Heifers

By Julien Soulat , Brigitte Picard * , Cécile Bord , Valérie Monteils

General Comment

This paper addresses the important study area of the effects of the rearing management on the carcass and meat properties of heifers. The paper addresses the topic in a way that has considerable practical implications and this is a great merit. The paper adds interesting information, addressing several aspects all of which are dealt with in a comprehensive manner. In general, the paper is relevant for the field and presented in a well-structured manner.

The authors however should make an effort to increase the number of refernces current with respect to those mentioned (more than 35% of the references are more than 5 years)

The paper need a minor revision.

General remarks:

I did not find any general remarks to make. Only one thing, change the Mcal in Mjaule

Detailed remarks:

Line 118: Color measurements: please, add a few more details..e.g. what kind of illuminant, were colour measurements made directly on the meat surface?..

Line 165: why was the shear force measured only for SV samples?

Line 346: “carcasses with a similar properties. “…not clear, please specify which properties.

Line 362: “The color of the cut section was significantly more homogeneous for heifers 362 performing the RM-2 than those performing the RM-3” incorrect

Line 367: “produced a darker meat (lower a* value)” seems unfair..explain better

Lines 398-399: “These both managements (RM1 and RM3) allowed to produce carcasses with smooth RH meat grain, a rib section with a more homogeneous color, a darker LM, and a yellower fat” the more homogeneous color is not correct (being RM2 not statistical different..).. and the yellower fat.. probably it is my fault but I do not understand from where it should be seen..it should perhaps be explained better

Table 1, 2 and 3: insert next to “rearing manegements”, RM in brackets, as done in table 4

Regards

Author Response

We would like to thank the reviewers for the relevance of their comments. The suggestions were very useful to improve the paper. We have addressed our responses to the reviewers below.

General Comment

This paper addresses the important study area of the effects of the rearing management on the carcass and meat properties of heifers. The paper addresses the topic in a way that has considerable practical implications and this is a great merit. The paper adds interesting information, addressing several aspects all of which are dealt with in a comprehensive manner. In general, the paper is relevant for the field and presented in a well-structured manner.

Thank you for your comments on our manuscript.

The authors however should make an effort to increase the number of refernces current with respect to those mentioned (more than 35% of the references are more than 5 years)

We had performed a literature review during the manuscript writing. Unfortunately, few current references are related to the aims of our study on the heifers. The study of rearing factors applied throughout the life of the animal is a specificity of the work carried out by the research team explaining a certain number of self-citations. We think that the references used are the most suitable with our topic and results.

 The paper need a minor revision.

 General remarks:

I did not find any general remarks to make. Only one thing, change the Mcal in Mjaule

We have changed the Mcal in kJ in the Tables 1-3

Detailed remarks:

Line 118: Color measurements: please, add a few more details..e.g. what kind of illuminant, were colour measurements made directly on the meat surface?..

We added more elements in the color assessment section (lines 128-131). Yes, the color was directly measured on the meat surface of each LM sample. We added also this information in the text (lines 130 -131).

Line 165: why was the shear force measured only for SV samples?

In fact, not all analysis were performed on the LM because the amount of LM was not sufficient to do that. In our study, as the LM is the reference muscle, we had prioritized the sensory and texture analysis on this muscle. As the rib is composed of many muscles, we have chosen to study the serratus ventralis muscle (specific muscle of the ribs in the chuck sale section). The aim was characterized more accurately the rib.

We added element in the text concerning the choice of muscles (lines 114-118).

Line 346: “carcasses with a similar properties. “…not clear, please specify which properties.

We have precised in the text the carcass traits which can be similar when the rearing management is different (lines 324).

Line 362: “The color of the cut section was significantly more homogeneous for heifers 362 performing the RM-2 than those performing the RM-3” incorrect

We apologize; there was a mistake on our part. In this sentence, the correct RM is RM-4. We carried out the modification in the text (line 340).

Line 367: “produced a darker meat (lower a* value)” seems unfair..explain better

We have modified the sentence. We have replaced “darker meat” by “less redness” in the text because the a* value represent the redness in the CIE color system (lines 343-344).

Lines 398-399: “These both managements (RM1 and RM3) allowed to produce carcasses with smooth RH meat grain, a rib section with a more homogeneous color, a darker LM, and a yellower fat” the more homogeneous color is not correct (being RM2 not statistical different..).. and the yellower fat.. probably it is my fault but I do not understand from where it should be seen..it should perhaps be explained better

We have rewritten the end of this section to better explain the idea that it was difficult to favor a rearing management and that trade-off were necessary (lines 378-382).

Table 1, 2 and 3: insert next to “rearing manegements”, RM in brackets, as done in table 4

We added “RM in brackets” in the title of the tables 1-3.

Reviewer 2 Report

As attachment

Author Response

We would like to thank the reviewers for the relevance of their comments. The suggestions were very useful to improve the paper. We have addressed our responses to the reviewers below.

Manuscript title:

Influence of the Rearing Managements on the Qualities of Carcass and Beef Meat in Charolais Heifers

General Comments

  • The manuscript is considering characterization of four rearing management systems and their impacts on carcass and meat qualities of Charolais heifers.
  • The topic of the manuscript is within the scope and aims of the journal.
  • The whole manuscript needs writing and grammatical revision for better and ease of understanding.

The manuscript was carefully reviewed and many changes were made throughout the text to improve understanding and correct grammatical and writing errors.

  • Throughout the manuscript, when using the word beef, no need to be followed by the word meat. It’s self-defined.

We have modified in all text.

  • At first sight, the reader will be confused whether the rearing system is an experimental factor or a target to be defined? Later, the rearing systems are characterized through the collected data and being used as experimental units. A clear and sharp definition of the rearing system is inevitable.

In our study, the rearing managements were characterized from the data collected during the surveys and performing a statistical analyze as described by Soulat et al. (2018). Then, in the analyses to observe the effect of the rearing management (RM) on the carcass and beef quality, the RM was considered as a factor.

We added a definition of the rearing management (lines 49-52) and we specified aims of the study (lines 55-57) in the introduction section.

Soulat, J., Picard, B., Léger, S., Ellies-Oury, M.-P., & Monteils, V. (2018). Preliminary Study to Determinate the Effect of the Rearing Managements Applied during Heifers’ Whole Life on Carcass and Flank Steak Quality. Foods, 7(10), 160.

I guess one of the main objectives of the study is to describe the four rearing systems. This should be stated clearly.

We have rewritten the aims of our paper to explain precisely the first aim, which was to characterize the rearing managements defined by statistical analyze, and studied in our work. (lines 55-57).

Title: I suggest a formative modified title:

Characterization of four rearing managements and their influence on carcass and meat qualities of Charolais heifers

Thank you for this proposal of title. It has been included in the manuscript (lines 2-3).

Abstract: Fair, but it’s recommended avoiding abbreviations in this section.

We have removed the abbreviation “RM” for the rearing management. However, the abbreviation for the longissimus muscle (LM) has been kept since it is usually used in the publications.

Introduction: Acceptable

Material and methods

  • This part needs extensive details of methodology regarding the set of experiment and evaluated parameters. It will be more reliable describing your methodologies in details rather than referring to tables.

We added elements to explain our methodology in the “material and methods” section. (lines 71-76, 89-91, 103, 128-131, 151-152, 174-175).

We think that including the definition of each variable in tables helps the understanding when reading of tables. Moreover, the presentation of the variable definitions in the tables allows to refer to the tables and to have an easier-to-read the “Material and methods” section and avoids redundancies.

  • A brief of the survey and main features of each rearing system has to be added in this section for the benefit of readers.

We added element to explain the data collected during the survey (lines 71-76).

The aims of our study were to characterize the rearing management applied during the heifer’s whole life, then to identify the effect of these rearing managements on the carcass and meat properties (lines 55-57).

As the rearing managements defined in this study are a result, it is not possible to explain these rearing managements in the “material and methods” section. In fact, the rearing managements were defined using hierarchical clustering from the rearing factors calculated from the batch management practices.

To help the readers, we have summarized the rearing managements defined in this study in the figure 2 and the paragraph lines 301-311).

  • Please, identify the total number of slaughtered animals and that selected from each rearing system.

We added the number of heifers slaughtered in the 2.2. section (line 87).

There was no selection of heifers within a rearing management. All data of rearing factors were used to define the rearing management using hierarchical clustering. Each heifer was related to a single rearing management explaining the differences in the number of heifers between the 4 rearing managements defined and considered in our study.

Results and Discussion

To consider the point 3.1 Description of the rearing managements defined as a result, the objective of the study, methodology and the title of the manuscript must be changed. Or this section has to be moved to material and methods part.

We have modified the title (lines 2-3) and rewritten the aims of our work (lines 55-57). As the rearing managements were statistically defined, their characterization is the first result of this work.

Line 183: Is MR-1 was performed by 42 heifers or 44 as appear in tables?

We apologize, there was a mistake. There was 44 heifers following the rearing management RM-1. We had modified this in the text (line 204).

Conclusions: Acceptable

References: Satisfied

Reviewer 3 Report

The objective of this study was conducted to increasing knowledge to improve the management of heifers and achieve better carcass and meat quality. The paper is interesting, although the item is not new. The paper contains a lot of information and there is an extensive description of the results. However, the discussion is carried out with many self-citations (similar works carried out by the research team) and with few citations from other researchers. In addition, more than an explanation of the results obtained, it is often only a coincidence or not with the results of other works.

Also, some specific fixes:

The affiliation of the last author is missing.

L80. Indicate the European regulations on which animal welfare in slaughterhouses is based.

L90. It is said that "11 carcass traits were assessed", for each rearing management?

L100. It is noted that “a sample of two beef ribs was collected”. Again, it refers to two ribs for rearing managements? Nor is it indicated how these samples were chosen.

L111. Indicate the color system used, based on the coordinates that are cited.

Information on the number of aftershocks on which the shear force measurements are carried out is missing.

In the results and discussion section, in section 2.3.2. Sensory Analysis, it is necessary to specify the number of sessions that have been carried out.

In the tables presented, it is advisable to present the global mean of all the animals, prior to the data of the different clusters.

It is suggested to synthesize the description of the results and make a deeper discussion of them.

Author Response

We would like to thank the reviewers for the relevance of their comments. The suggestions were very useful to improve the paper. We have addressed our responses to the reviewers below.

Comments and Suggestions for Authors

The objective of this study was conducted to increasing knowledge to improve the management of heifers and achieve better carcass and meat quality. The paper is interesting, although the item is not new. The paper contains a lot of information and there is an extensive description of the results. However, the discussion is carried out with many self-citations (similar works carried out by the research team) and with few citations from other researchers. In addition, more than an explanation of the results obtained, it is often only a coincidence or not with the results of other works.

Thank you for your comments. We agree with your comment concerning the citations used. However, we have performed a literature review during the manuscript writing. We have not found many publications on the heifers and considering the whole life of the animal. The study of rearing factors applied throughout the life of the animal is a specificity of the work carried out by the research team, which explains a certain number of self-citations. We think that the references used are the most suitable with our topic and results.

Considering the specificity of the heifer’s whole life study, it was difficult to compare our results with previous published works generally carried out only on the fattening period. Moreover few data were available in the literature on the heifers. When it was possible comparisons were done with individual rearing factors. We added elements in the manuscript to explain this (lines 345-346, 373-375, 395-397, 438-442, 459-460).

Also, some specific fixes:

The affiliation of the last author is missing.

We added the affiliation of the last author (line 4)

L80. Indicate the European regulations on which animal welfare in slaughterhouses is based.

We indicate the European regulations in the text concerning “the protection of animals at the time of killing” ‘(No 1099/2009) (lines 89-91).

L90. It is said that "11 carcass traits were assessed", for each rearing management?

Irrespective of the rearing managements, the 11 carcass traits described in the Table 4 were assessed on the carcasses in the slaughterhouse. Then, using ANOVA, we observed the effect of the rearing managements on all carcass traits.

L100. It is noted that “a sample of two beef ribs was collected”. Again, it refers to two ribs for rearing managements? Nor is it indicated how these samples were chosen.

Irrespective of the rearing managements, two ribs were collected in the chuck sale section for each carcass. These samples were chosen because the ribs contained the longissimus muscle (LM, the most studied muscle in the published studies, i.e. reference muscle). Moreover, this sample limits the depreciation of the carcasses for the further sale.

L111. Indicate the color system used, based on the coordinates that are cited.

We added the color system used in the text (lines 128 and 129).

Information on the number of aftershocks on which the shear force measurements are carried out is missing.

We added a sentence in the text to precise the total number of shear force measurements performed per SV sample. (lines 174 – 175).

In the results and discussion section, in section 2.3.2. Sensory Analysis, it is necessary to specify the number of sessions that have been carried out.

We added the number of sessions in the section 2.3.3. (lines 151-152). In the results and discussion section, we think it is not necessary to precise the number of sensory sessions.

In the tables presented, it is advisable to present the global mean of all the animals, prior to the data of the different clusters.

We added the global mean of all animals in all tables.

It is suggested to synthesize the description of the results and make a deeper discussion of them.

We have tried to synthesize the description of the results for easy reading. However, as there are many rearing factors, the description of the rearing managements is still important.

Concerning the discussion, we added new elements in the “results and discussion” section. However, the difficulty was to compare our results with previous published results outside our own. In our study, we worked on the combination of many rearing factors to study the rearing management applied during the heifer’s whole life. However, there are few published results in heifers. Then, the publications are generally focus on the fattening period and studied the individual effect of rearing factor. Comparisons of our results with the literature were made when possible.

Reviewer 4 Report

This work has a good scientific quality and is well designed, conducted, and reported, the results were argued well. However, need to revise some questions. I recommend a minor revision. There are spelling and grammatical errors throughout the manuscript, some of which have been pointed out in the specific comments. There are areas of the manuscript where English grammar could be improved. I suggest reviewing the conclusions, remembering that they must be supported by the results shown in the manuscript, and giving a response to the hypothesis. In my opinion, in addition to the existing ones, another keyword should be inserted.

Comments to authors (e.g. suggestions of changes to the text):

Line 12: …171 heifers selected into commercial… in this part of the sentence, replace into with for is more suitable

Line 19: … produced had a high…

Line 19:… LM meat were more liked?

Line 27: … In 2020, meat production ‘’in’’ the European Union

Line 33: However, ‘for’’ many years…

Line 37: … the quantity and to increase…

Line 42: stays factor?

Line 44: …by many factors as e.g. rearing factors…

Line 68: … produced on the farm as e.g. barley…

Line 80: The heifers were stunned using a captive-bolt…

Line 82: which term is more suitable, weighted or weighed?

Line 83: … according ‘’to’’ the EUROP…

Line 93: …the Figure 1 using…

Line 96: homogeneity

Line 104: ... were thawed for around…

Line 112: … area of LM as the visual color…

Line 120: … perceptions and use of the perception…

Line 140: … thawed ‘’for’’ 25 min…

Line 150: … at +4 °C for around 24h.

Line 153: … using the Warner-Braztler…

Line 164: … HCPC allowed to determining the number…

Line 168: … each rearing factor

Line 187: … The calves did not receive concentrates… and the same mistake in the next line

Line 192: … heifers from the both managements…

Line 195: … did not receive corn silage… and the same mistake in the line 199

Line 218: … calves received ‘’an’’ average…

Line 345: … did not observe a…

Line 347: … also similar ‘’to’’ rearing managements…

Line 351: … mainly based ‘’on’’ corn silage…

Line 355: … Duckett et al. [47] ‘’observed’’ that the fat…

Line 363: … had no significant…

Line 371: … However, the effect of the both managements…

Line 387: … than the other rearing…

Line 390: … Moloney et al. [58] did not observe an effect…

Line 393: … raw meats were not significantly…

Line 397: … did not observe an effect…

Line 412: … to produce an LM meat…

Line 423: … In a previous study…

Line 426: … the both...

Line 433: … to produce ‘’of’’ carcasses with ‘’a’’ high…

Author Response

We would like to thank the reviewers for the relevance of their comments. The suggestions were very useful to improve the paper. We have addressed our responses to the reviewers below.

Comments and Suggestions for Authors

This work has a good scientific quality and is well designed, conducted, and reported, the results were argued well. However, need to revise some questions. I recommend a minor revision.

Thank you for your comments

There are spelling and grammatical errors throughout the manuscript, some of which have been pointed out in the specific comments. There are areas of the manuscript where English grammar could be improved.

The manuscript was carefully reviewed and many changes were made throughout the text to improve understanding and correct grammatical and writing errors.

I suggest reviewing the conclusions, remembering that they must be supported by the results shown in the manuscript, and giving a response to the hypothesis.

The conclusion section was changed (lines 452-460). We added elements related to the aims developed in the introduction section. The hypothesis of an effect of the rearing management (from birth to slaughter) on the carcass and meat quality was confirmed. Then, it was possible to manage simultaneously these both qualities.

In my opinion, in addition to the existing ones, another keyword should be inserted.

We added 3 new keywords in the manuscript (line 24).

Comments to authors (e.g. suggestions of changes to the text):

Line 12: …171 heifers selected into commercial… in this part of the sentence, replace into with for is more suitable

We have modified the sentence (line 12-13)

Line 19: … produced had a high…

We carried out the modification in the text (line 20)

Line 19:… LM meat were more liked?

The LM meat from heifers performing the RM-3 had a significant high overall acceptability value compared to the LM meat from the rearing managements RM-1 and RM-2. The “more liked” refer to this hedonic descriptor.

Line 27: … In 2020, meat production ‘’in’’ the European Union

We carried out the modification in the text (line 29)

Line 33: However, ‘’for’’ many years…

We carried out the modification in the text (line 35)

Line 37: … the quantity and to increase…

We carried out the modification in the text (line 39)

Line 42: stays factor?

We think that “stay” have not “s” because the subject is “the sensory properties”. Then, as there are many sensory descriptors, we think that “factor” is plural form.

Line 44: …by many factors as e.g. rearing factors…

We carried out the modification in the text (line 45)

Line 68: … produced on the farm as e.g. barley…

We carried out the modification in the text (line 78)

Line 80: The heifers were stunned using a captive-bolt…

We carried out the modification in the text (line 92)

Line 82: which term is more suitable, weighted or weighed?

We carried out the modification in the text (line 94). The right word was “weighed”.

Line 83: … according ‘’to’’ the EUROP…

We carried out the modification in the text (line 95)

Line 93: …the Figure 1 using…

We carried out the modification in the text (line 105)

Line 96: homogeneity

We carried out the modification in the text (lines 108, 314, and Table 4). We had used “homogeneous color” to replace “color homogeneity”.

Line 104: ... were thawed for around…

We carried out the modification in the text (line 120)

Line 112: … area of LM as the visual color…

We carried out the modification in the text (line 131)

Line 120: … perceptions and use of the perception…

We carried out the modification in the text (line 139)

Line 140: … thawed ‘’for’’ 25 min…

We carried out the modification in the text (line 160)

Line 150: … at +4 °C for around 24h.

We carried out the modification in the text (line 170)

Line 153: … using the Warner-Braztler…

We carried out the modification in the text (line 172)

Line 164: … HCPC allowed to determining the number…

We carried out the modification in the text (line 185)

Line 168: … each rearing factor

We carried out the modification in the text (line 189)

Line 187: … The calves did not receive concentrates… and the same mistake in the next line

We carried out the modification in the text (lines 208 and 209)

Line 192: … heifers from the both managements…

We removed “both managements” (line 222)

Line 195: … did not receive corn silage… and the same mistake in the line 199

We carried out the modification in the text (lines 215 and 219)

Line 218: … calves received ‘’an’’ average…

We carried out the modification in the text (line 236)

Line 345: … did not observe a…

We carried out the modification in the text (line 354)

Line 347: … also similar ‘’to’’ rearing managements…

We carried out the modification in the text (line 355)

Line 351: … mainly based ‘’on’’ corn silage…

We carried out the modification in the text (line 359)

Line 355: … Duckett et al. [47] ‘’observed’’ that the fat…

We carried out the modification in the text (line 363)

Line 363: … had no significant…

We carried out the modification in the text (line 371)

Line 371: … However, the effect of the both managements…

We removed “both managements” (line 378)

Line 387: … than the other rearing…

We carried out the modification in the text (line 399)

Line 390: … Moloney et al. [58] did not observe an effect…

We carried out the modification in the text (line 402)

Line 393: … raw meats were not significantly…

We carried out the modification in the text (line 404)

Line 397: … did not observe an effect…

We carried out the modification in the text (line 407)

Line 412: … to produce an LM meat…

We think it is not necessary to use “an” because it is not preceded by a vowel.

Line 423: … In a previous study…

We carried out the modification in the text (line 430)

Line 426: … the both...

We removed “both managements” (line 434)

Line 433: … to produce ‘’of’’ carcasses with ‘’a’’ high…

We carried out the modification in the text (line 445)
